# Pharmacogenetics of Osteoporosis: A Pathway Analysis of the Genetic Influence on the Effects of Antiresorptive Drugs

**DOI:** 10.3390/pharmaceutics14040776

**Published:** 2022-04-02

**Authors:** Álvaro del Real, Carmen Valero, José M. Olmos, Jose L. Hernández, José A. Riancho

**Affiliations:** 1Departamento de Medicina y Psiquiatría, Universidad de Cantabria, 39008 Santander, Spain; delreala@unican.es (Á.d.R.); carmen.valero@unican.es (C.V.); manuel.olmos@unican.es (J.M.O.); joseluis.hernandez@unican.es (J.L.H.); 2Servicio de Medicina Interna, Hospital U.M. Valdecilla, Instituto de Investigación Sanitaria Valdecilla (IDIVAL), 39008 Santander, Spain

**Keywords:** pharmacogenomics, antiresorptives, osteoporosis

## Abstract

Osteoporosis is a skeletal disorder defined by a decreased bone mineral density (BMD) and an increased susceptibility to fractures. Bisphosphonates and selective oestrogen receptor modulators (SERM) are among the most widely used drugs. They inhibit bone resorption by targeting the mevalonate and oestrogen pathways, respectively. The aim of this study was to determine if common variants of genes in those pathways influence drug responses. We studied 192 women treated with oral aminobisphosphonates and 51 with SERMs. Genotypes at 154 SNPs of the mevalonate pathway and 806 in the oestrogen pathway were analyzed. Several SNPs located in genes FDPS and FNTA were associated with the bisphosphonate-induced changes in hip bone mineral density (BMD), whereas polymorphisms of the PDSS1, CYP19A1, CYP1A1, and CYP1A2 genes were associated with SERM-induced changes in spine BMD. After multivariate analyses, genotypes combining genes FDPS and FNTA showed a stronger association with bisphosphonate response (r = 0.34; *p* = 0.00009), whereas the combination of CYP19A1 and PDSS1 genotypes was associated with the response to SERMs (r = 0.62, *p* = 0.0003). These results suggest that genotyping genes in these pathways may help predict the response to antiresorptive drugs and hence make personalized therapeutic choices.

## 1. Introduction

Osteoporosis is a highly prevalent disorder, particularly among postmenopausal women and elderly men. It is characterized by a decrease in bone mass and alterations of bone microarchitecture that impair the ability of the skeleton to resist mechanical loads [1]. Fragility fractures, including those at the spine and the hip among others, are the clinically relevant consequence of osteoporosis. Among factors influencing bone strength, bone mineral density (BMD) is the most relevant from a clinical point of view. In fact, there is an exponential relationship between BMD and fracture risk. Therefore, BMD, assessed by dual-energy X-ray absorptiometry (DXA) at the lumbar spine and the hip, is used in clinical practice to diagnose osteoporosis as well as to estimate fracture risk, along with some other clinical factors.

Osteoporosis, and specifically osteoporosis-related fractures, represent an important burden both for individual patients and the health systems. Fortunately, several drugs effectively impact BMD and decrease fracture risk. The most widely used drugs belong to the antiresorptive class because they inhibit bone resorption. Among them, selective oestrogen receptor modulators (SERMs) increase BMD at the spine and lower the risk of vertebral fractures but do not show consistent effects at the hip [2]. Aminobisphosphonates are also widely used antiresorptive drugs that target the mevalonate pathway and decrease the risk of both vertebral and nonvertebral fractures [3]. Antiresorptive agents are usually well-tolerated and effective for many patients, but they are not universally effective. In fact, some patients do not experience an increase in BMD and/or may sustain fragility fractures despite therapy. For instance, the relative risk reduction attained with oral bisphosphonates (BPs), such as alendronate or risedronate, is roughly 40–60% and 20–40%, for vertebral and peripheral fractures, respectively [4,5,6].

Bone turnover is a major determinant of BMD, which is the consequence of the balance between bone formation and bone resorption. These processes take several months at each remodelling site. Therefore, therapy-induced changes in BMD can provide useful information about drug effectiveness, but usually take at least 2 years to develop. Hence, for clinicians and patients making therapeutic choices, it could be very interesting to have tools helping to predict the probability of responses to a certain drug.

Bone mass has an important hereditary component [7]. Genetic association studies, such as candidate gene and genome-wide studies (GWAS) have identified gene variants associated with BMD and fracture risk, including some in genes known to influence skeletal homeostasis, such as those of the oestrogen and Wnt pathways [8,9,10,11,12]. Hence, we planned this study to determine if common allelic variants of the mevalonate pathway and the oestrogen pathway influenced the response to BPs and SERMs, respectively.

## 2. Materials and Methods

### 2.1. Patients and Densitometry

We included postmenopausal osteoporotic women who were treated with oral aminobisphosphonates (alendronate or risedronate; mean age, 67 years; range 47 to 87 years) or with SERMs (raloxifene or bazedoxifene, mean age 57 years; range, 47 to 77 years). Subjects diagnosed with secondary osteoporosis were excluded. BMD was measured twice by dual X-ray densitometry (DXA) at the spine and the hip, both at baseline and after 1 to 4 years of treatment, using a Hologic QDR 4500 densitometer (Waltham, MA). In vivo precision was 0.5% at the lumbar spine and 0.4% at total hip. The mean annual change in BMD was computed for each patient and region and expressed as the percentage of baseline BMD. Patients with annual BMD changes >6% or <−6% were regarded as unreliable, and thus, they were excluded from the analysis.

The study protocol was approved by the institutional review board (Comité de Ética en Investigación Clínica de Cantabria). All patients gave informed written consent.

### 2.2. DNA Isolation and Genotyping

DNA was isolated from 200 µL aliquots of peripheral blood using commercially available column-based kits, following manufacturer instructions. DNA was quantitated by both Nanodrop and Qubit dsDNA BR Assay Kit (ThermoFisher, Waltham, MA, USA).

DNA samples were genotyped at the Spanish National Genotyping Center (CeGen-PRB3-ISCIII; Santiago de Compostela, Spain), using the Axiom™ Spain Biobank array and strictly following the manufacturer’s instructions (Axiom™ 2.0 Assay 96-Array Format Manual Workflow; ThermoFisher Scientific). Briefly, total genomic DNA (200 ng) was amplified and randomly fragmented into 25 to 125 base pair fragments, which were then purified and resuspended in a hybridization cocktail. The hyb-ready targets were then transferred to the GeneTitan Multichannel Instrument for automated, hands-free processing (including hybridization to Axiom array plates, staining, washing and imaging). CEL files were automatically processed for allele calling using the Axiom GT1 algorithm available through the Axiom Analysis Suite v4.0.3.3 and following the Axiom™ Genotyping Solution Data Analysis User Guide (ThermoFisher Scientific, Waltham, MA, USA).

### 2.3. Quality Control and Data Analysis

The Axiom Analysis Suite and PLINK software were applied to conduct the quality control of genotype data. Thresholds were DQC: ≥0.82 and call rate: ≥97%. The percent of passing samples was ≥95, and the average call rate for passing samples was ≥98.5. In addition, SNPs with minor allele frequency less than 0.02, or departure from the Hardy-Weinberg equilibrium -HWE- (*p* < 1 × 10^−6^), were also excluded from further analyses.

SNPs associated with genes of the mevalonate and oestrogen pathways were identified and taken forward for association analysis. The association of individual SNPs with BMD changes was tested in linear models, with or without potential confounders (including body mass index, baseline BMD or the interval between DXA scans). These analyses as well as haplotype-based analysis were carried out using PLINK software [13]. The significance threshold was adjusted by the number of SNPs associated with each gene. The linkage disequilibrium (LD) of the significant probes within the same gene was studied with Haploview. Haplotypic blocks were estimated with the method of Gabriel [14].

Analyses combining alleles from several genes were carried out by multiple linear regression using the R package “olsrr”. Gene-based analysis that incorporates the association data of all SNPs in the gene was carried out with MAGMA [15].

## 3. Results

### 3.1. Patients, BMD Changes and SNPs

We included 243 patients in the analysis. As stated, those with BMD changes >6% were excluded because of potential errors in DXA analysis. Thus, the final study sample included 205 patients with valid densitometric data for the lumbar spine and 230 for the total hip. Table 1 shows the main baseline features of the study population. The individual mean annual changes in BMD are shown in Figure 1.

The mean annual change in BMD in BP-treated patients was 2.2% at the lumbar spine (*p* = 2.2 × 10^−16^) and 0.8% for the total hip (*p* = 9.5 × 10^−9^). In those on SERMs, the mean annual changes were 0.8% and 0.4%, at the spine (*p* = 0.0065) and the hip (*p* = 0.1225), respectively.

After quality control and excluding SNPs violating HWE and those with MAF < 0.02, 531,131 SNPs were used for the analysis, 154 related to the mevalonate pathway genes and 806 related to the oestrogen pathway. Further details are shown in Appendix A.

### 3.2. Single Locus Analysis

#### 3.2.1. Mevalonate Pathway

The association between BMD changes and 154 SNPs related to the 22 genes of the mevalonate pathway is shown in Figure 2. A total of 7 SNPs (related to genes FNTB, MVD, PDSS1, and PDSS2) showed a nominally significant association (*p* < 0.05) with the BP-induced change in spine BMD, whereas 15 SNPs (in FDPS, FNTA, FNTB, IDI1, MVD, MVK, PDSS1, and PDSS2 genes) were associated with hip BMD changes (Table 2). Additionally, 10 and 5 SNPs showed a nominally significant association with BMD changes at the spine and the hip, respectively, in SERM-treated patients. No SNP reached the multiple test-corrected significance threshold (0.05/154 = 0.00032).

#### 3.2.2. Oestrogen-Related Pathway

The association of the 806 SNPs related to the 36 genes in the oestrogen pathway with BMD changes in SERM-treated patients is shown in Figure 3. Overall, 49 SNPs (CYP19A1, CYP2D6, EGF, ESR1, ESR2, FOS, MAP2K1, MAPK1, MAPK8, NFKB1, UGT1A8, CYP1A1, CYP1A2, CYP3A4, and SULT1A1 genes) were associated with SERM-induced changes in spine BMD, and 50 SNPs were associated with hip BMD changes (CYP19A1, ESR1, ESR2, FOS, GPER1, JUN, MAP2K1, MAPK1, MAPK14, MAPK8, ARSD, ARSL, CYP1A1, CYP1A2, and CYP1B1 genes). For comparison purposes, the results in BP-treated patients are also shown. A total of 50 SNPs were nominally associated (*p* < 0.05) with spine BMD changes and 31 with hip BMD changes. None of them reached the overall-corrected significance threshold (0.05/806 = 0.000062).

The results were also explored using Venn diagrams (Figure 4), showing common and distinct SNPs associated with BMD changes according to the region studied and the treatment group.

### 3.3. Multimarker and Gene-Based Analysis

Since no significant changes in hip BMD were found in the SERM-treated group, these data were not further analysed. The number of SNPs included varied across the genes studied, and we used a significance threshold adjusted by the number of SNPs in each gene. Thus, genes with at least one SNP significantly associated with BMD changes are shown in Table 3. BMD changes across genotypes of some representative SNPs are shown in Figure 5.

#### 3.3.1. Haplotypic Analysis

In those genes with two significant SNPs (CYP19A1, CYP1A1, and FDPS), an omnibus haplotype-based association analysis was performed. In all three genes, a significant association with drug-induced change in BMD was detected (*p* = 0.0008; *p* = 0.0009, and *p* = 0.0065, respectively). The three pairs were in strong LD, with D’ score values equal to 1 (Figure 6). Haplotypes’ frequencies and association tests for each pair of variants are represented in Table 4.

#### 3.3.2. Gene-Based Analysis

For those genes with at least one significant SNP in the individual analysis, a gene-based analysis was carried out using multi-marker analysis of genomic annotation (MAGMA), with a window of 10 kb upstream and 10 kb downstream of each gene. The results are shown in Table 5. In BP-treated patients, there was a nominally significant association between FDPS variations and BMD changes at the hip and between PDSS1 and BMD changes at the spine. In the SERM group, the polymorphisms in both the CYP19A1 and CYP1A1 genes were associated with the treatment-dependent changes in spine BMD, with the former crossing the significance-adjusted threshold (0.05/6 = 0.008).

#### 3.3.3. Multivariate Analysis

Performing multivariate linear regression analyses, by selecting the most significant SNP of each gene and adjusting the results by anthropometric variables and duration of therapy (that is, the interval between the two DXA studies), did not significantly modify the results. There was no interaction between the variables.

We also explored the combined effect of several genetic polymorphisms using stepwise linear regression. The combination of FDPS and FNTA gene polymorphisms increased the association value of each separate gene with the hip BMD changes in the BP group (Table 6). There was no evidence of interaction among those predictors (*p* = 0.25). The mean changes in each genotype combination are shown in the Figure 7.

The combination of PDSS1, CYP19A1, CYP1A1, and CYP1A2 genes also increased the association value (in comparison with each gene separately) with spine BMD changes in the SERMs group. In the stepwise analysis, CYP19A1 and PDSS1 were the genes with a significant independent association with spine BMD change (Table 7). There was no evidence of interaction between them (*p* = 0.65). The average changes in each combined genotype are shown in Figure 7.

## 4. Discussion

### 4.1. Pharmacodynamics and Therapeutic Role of Antiresorptive Drugs

Bone mass and strength are maintained by the balance between bone resorption and bone formation, carried out by osteoclasts and osteoblasts, respectively. Thus, when bone resorption exceeds bone formation, bone mass decreases, and the strength of bone is impaired, leading to a high risk of fragility fractures, the main consequence of osteoporosis. Several factors determine the strength of the skeleton, including absolute bone mass, geometry, bone turnover, microarchitecture, and some other parameters of bone quality. Nevertheless, in clinical practice, BMD assessed by DXA is the main determinant of fracture risk and an important surrogate marker of the antifracture effect of antiosteoporotic drugs [16].

Antiresorptive agents, including BPs, SERMs, and denosumab, are the most widely used therapy for osteoporosis. The oral BPs, alendronate and risedronate, are the preferred therapy for many patients. These drugs inhibit bone resorption, reduce bone turnover, increase BMD, and decrease the risk of vertebral and nonvertebral fractures, including hip fractures [4].

AminoBPs, such as risedronate and alendronate, act by targeting the mevalonate pathway, which is involved in the biosynthesis of cholesterol and isoprenoid lipids [17]. The latter are needed for the prenylation of proteins, a process that involves the transfer of 15-carbon or 20-carbon isoprenoid groups onto a cysteine residue. This is mediated by prenyltransferases that can attach either a farnesyl group (15-carbon) or a geranylgeranyl group (20-carbon) in thioether linkage to the cysteine residue of many proteins. In eukaryotic cells, there are three protein prenyltransferases: farnesyltransferase and type I and type II geranylgeranyltransferases.

Many regulatory proteins are subjected to prenylation, including small GTPases of the Ras, Rho, Rac, Cdc42, and Rab families. They play a critical role in the regulation of osteoclast activity, and subsequently bone resorption, such as cytoskeletal arrangement, membrane ruffling, trafficking of intracellular vesicles, and apoptosis [17].

Type I geranylgeranyltransferase and farnesyltransferase are heterodimers that share the same alpha subunit (encoded by the FNTA gene) but have different beta subunits (encoded by the FNTB and PGGT1B genes, respectively). Type II geranylgeranyl transferase (also known as RabGGTase) chains are encoded by the RabGGTA and RabGGTB genes [18]. Farnesyltransferase selectively prenylates H-Ras, whereas Type I geranylgeranyltransferase acts on Rho, K-Ras, Rap, and Rab [19]. Proper prenylation of these proteins is required for osteoclast-mediated bone resorption. In addition, type II geranylgeranyltransferase may be involved in the apoptosis of some tumor cells [18].

Farnesyl diphosphate synthase (FDPS) is the main target of BPs, but they may also inhibit other enzymes, including isopentenyl diphosphate (IPP) isomerase, geranylgeranyl diphosphate synthase (GGPP), geranylgeranyl diphosphate synthase (GGDPS), geranylgeranyl transferase, and squalene synthase [17,18]. Interestingly, upregulation of endogenous levels of FDPS confers partial resistance to BPs [20].

### 4.2. Relationship of Gene Variants with the Effect of Bisphosphonates

In this study, we found an association between the changes in hip BMD induced by BP and allelic variants of the FDPS and FNTA genes, encoding FDPS and the alpha chain of farnesyl transferase, respectively (see Figure 8). The association between FDPS polymorphisms and BP response confirms previous results from several groups, including our own, and is in line with the concept of FDPS as a major protein target of BP [8]. In the present analysis, three different SNPs located on the FDPS gene showed an association with hip BMD changes induced by oral BP therapy. These SNPs, rs2297480, rs11264359, and rs10908463, are located in the first, second, and third introns, respectively. All three are in strong LD with a D’ score equal to 1. The result is in line with previous studies, in which various groups found a smaller BMD response to BP in women with the minor allele of rs2297480 or rs11264359 (the alternative allele) [21,22,23,24,25]. The alleles of these three SNPs are associated with FDPS expression (GTEX portal: https://gtexportal.org (accessed on 14 February 2022)), with normalized effect sizes of about 0.3. Hence, it is unclear if any of these SNPs or some others else in linkage disequilibrium is leading the pharmacogenetic effect.

Our results also indicate that FNTA genetic variants are related to the BP effect. In particular, rs35601968, a noncoding variant located in the first intron of the FNTA gene. The minor allele of rs35601968 was also associated with less positive changes in BMD following BP therapy. This SNP is located about 27 kb from the TSS, and no evidence supports its role as an eQTL. Therefore, the mechanisms explaining the association are unknown. It is unclear if prenyltransferases are direct targets of BPs. However, the relative abundance of farnesyltransferases and geranylgeranyltransferases impact the balance between farnesylation and geranylgeranylation of proteins, which in turn may influence cellular activities [26]. Therefore, allelic variants influencing FNTA expression may likely influence the activity of osteoclasts and their response to BPs.

### 4.3. Relationship of Gene Variants with the Effect of SERMs

Bone remodelling and bone turnover are under the control of several local and systemic factors. Among them, oestrogens play a major role in bone mass acquisition and maintenance. Thus, oestrogens decrease bone resorption and reduce the risk of fractures. Unfortunately, the possibility of adverse effects precludes widespread use of these agents for preventing or treating osteoporosis. SERMs were developed to overcome some of these drawbacks, particularly the protumorigenic effect. Thus, SERMs, such as raloxifene or bazedoxifene, have a neutral or inhibitory effect on the proliferation of endometrial and breast tissues, while they increase BMD at the spine and lower the risk of vertebral fractures. However, they have a smaller antiresorptive effect than oestrogens, and they do not have clear effects on bone mass at the hip [2]. In line with this concept, we found that SERMs induced a significant increase of BMD at the spine but not at the hip.

Ovaries are the main source of oestrogens in young women. However, in postmenopausal women and men, a certain level of oestrogens is maintained by the activity of aromatase (Figure 8). This enzyme, encoded by the CYP19A1 gene, converts androgenic precursors into oestrogens in the peripheral tissues. In fact, allelic variants of the CYP19A1 gene are associated with bone mass in late postmenopausal women and men but not in younger women [27,28,29].

In this study, we found that some common variants in the CYP19A1 gene are associated with the response to SERM therapy. Those SNPs are in close LD, and the SNP driving the association with drug response is still unclear. However, by using several procedures, such as quantitative PCR analysis of the natural transcript abundance, allele-specific expression or expression of fragments cloned into reporter vectors, we have previously shown that those SNPs are associated with relevant differences in aromatase gene expression that are translated into the circulating levels of oestradiol [27,29]. A total of 10 CYP19A1 SNPs were nominally associated (i.e., *p* < 0.05) with lumbar spine BMD changes after SERM therapy. Two of them, rs3889391 and rs1062033, showed *p*-values below the adjusted threshold, considering the number of SNPs analysed in the CYP19A1 gene. Both intronic variants were in strong LD, and their minor alleles were associated with a smaller BMD response induced by SERMs treatment. These SNPs are also in strong LD with other SNPs located from the 5’-region down to the coding region of CYP19A1 (Figure 9) and are associated with the abundance of aromatase. Interestingly, women bearing alleles associated with more active gene transcription according to the GTEX portal and previous data from our laboratory had minor BMD responses to SERM. It is tempting to speculate that this was related to their somewhat higher endogenous oestrogen levels.

PDSS1 encodes the subunit 1 of Decaprenyl Diphosphate Synthase. This enzyme produces prenyl diphosphates of varying chain lengths and elongates the prenyl side chain of coenzyme Q, or ubiquinone, a component of the respiratory chain. We found an association of SERM-induced changes in BMD with allelic variants of PDSS1, specifically with rs11015261, located in the last intron of the gene, with higher BMD responses in women bearing the minor allele. The exact biological role of PDSS1 is unclear, but the expression of PDSS1 has been associated with the breast cancer response to the first-generation SERM, tamoxifen [30]. Whatever the involved mechanisms might be, they could perhaps also be involved in the association of PDSS1 alleles with the bone response to SERMs. However, we also found a trend for an association of PDSS1 and PDSS2 alleles with the response to BP. Thus, it is unclear whether or not the association is drug-specific. Nevertheless, coenzyme Q may modulate osteoclast differentiation [31]. Mitochondrial dysfunction alters the activity of osteoclasts, osteoblasts, and their precursor stem cells and may be involved in the bone loss associated with aging and oestrogen deficiency [32].

### 4.4. Clinical Relevance

Bouxsein et al. estimated that for a 2% or 6% improvement in total hip BMD, we might expect a 16% or 40% reduction in hip fracture risk [16]. According to the mean annual changes in BMD observed after antiresorptive therapy, the pharmacogenetic effect sizes found in this study may be clinically relevant. For example, women bearing the GT genotype at the rs2297480 locus of FDPS who were on BP gained about one-third of the BMD gained by those with the most common TT genotype (mean changes 0.39 and 1.07, respectively), and the 3% of women with the least common GG genotype did not experience any increase in BMD. Similar relationships were found across CYP19A1 genotypes in women treated with SERMs.

The GWAS of osteoporosis-related outcomes and other complex traits and disorders have shown a small contribution of each individual SNP to the phenotype [11]. Hence, there is an interest in combining several polymorphisms into polygenic risk scores. In this sense, several genetic scores have been built that somewhat add to the prediction of calcaneal ultrasound parameters, BMD, or fractures [33,34,35]. In the same line, combining the allelic variants of two genes, FDPS and FNTA, explained 11.6% of the variance of the BP-induced change in BMD, whereas each separate gene explained 4.4 and 6.8%, respectively. Similarly, CYP19A1 and PDSS1 explained 23% and 22.1% of the variance of the BMD change in response to SERMs, and the variance explained increased up to 38.4% when both genes were considered in combination. If these figures are confirmed in large independent cohorts, thus confirming the generalizability of these results, knowing the alleles at those loci could be useful to make therapeutic decisions in patients with low bone mass.

### 4.5. Study Limitations

Our study has several limitations. We included a well-characterized and homogenous group of postmenopausal women. All were of Spanish ancestry and had primary osteoporosis, reducing bias introduced by differences in genetic background or environmental factors. However, the sample size was small. Therefore, although plausible because of their biological rationale and in line with other literature data, we cannot completely exclude type I error inflation in some associations. Thus, the study should be considered exploratory, and the results should be confirmed in larger cohorts of patients. Moreover, the sample size limited the statistical power. Hence, it is likely that, besides those pointed out in this study, some other genes may influence the response to antiresorptive drugs. In addition, we do not know whether any of these allelic variants are associated with the response to other antiresorptive agents, such as denosumab or anabolic drugs such as teriparatide, abaloparatide or romosozumab. We used BMD changes as the measure of drug effect. However, fractures are the clinically relevant consequence of osteoporosis. Although BMD increase following drug therapy is clearly associated with the reduced risk of fractures, DXA changes do not capture the whole anti-fracture efficacy. Very large cohorts, including thousands of patients, are likely needed to reveal the potential effects of allelic variants on fracture occurrence while on therapy. As an alternative approach in future studies, it could be interesting to correlate the genotypic variants with the drug-induced changes in bone turnover markers, as they are also associated with the decrease in the risk of vertebral fractures attained with antiresorptive agents [36,37].

## 5. Conclusions

In summary, our results suggest that common allelic variants in genes involved in the mevalonate pathway, such as FDPS and FNTA, are associated with the BMD response to BP. Variants in the CYP19A1 (aromatase) gene, required for oestrogen synthesis, particularly after menopause, and in the PDSS1 gene, which is part of the mitochondrial respiratory chain, are associated with the response to SERMs. These results may help build polygenic scores that help predict the response to drug therapy and therefore, to personalize therapy of osteoporosis and other bone fragility conditions. On the other hand, in view of these results, it may be worthwhile to explore the skeletal effect of drugs targeting mevalonate-related enzymes other than bisphosphonates.

## Figures and Tables

**Figure 1 pharmaceutics-14-00776-f001:**
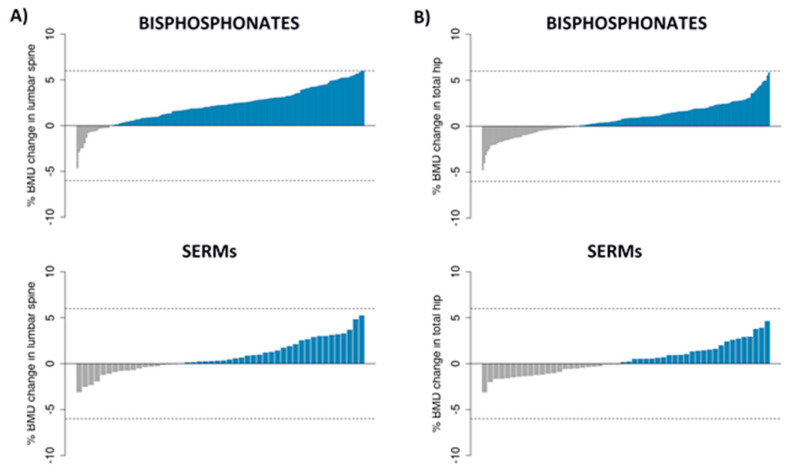
Distribution of the BMD change (as percent of baseline) in each region analyzed: (**A**) lumbar spine; and (**B**) total hip. Dotted lines are the limits for an absolute change value greater than 6%.

**Figure 2 pharmaceutics-14-00776-f002:**
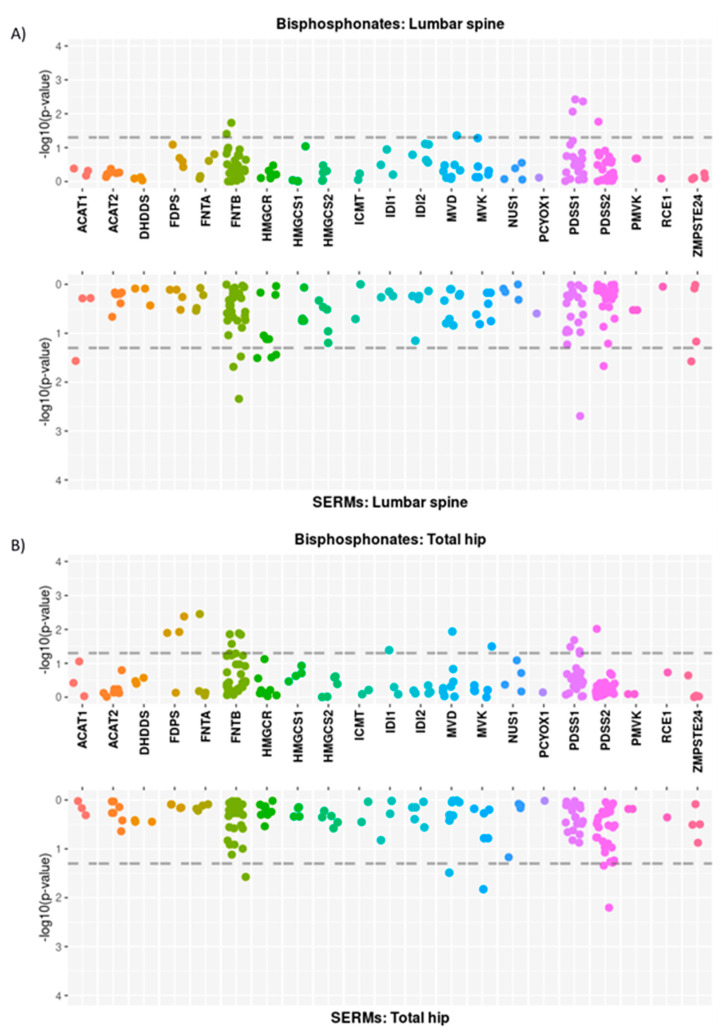
Miami plots showing the *p*-values of the association between mevalonate pathway SNPs and annual BMD change at the spine (**A**) and hip (**B**). For both regions, the results of the BP-treated group (upward) and the SERM-treated groups (downward) are shown.

**Figure 3 pharmaceutics-14-00776-f003:**
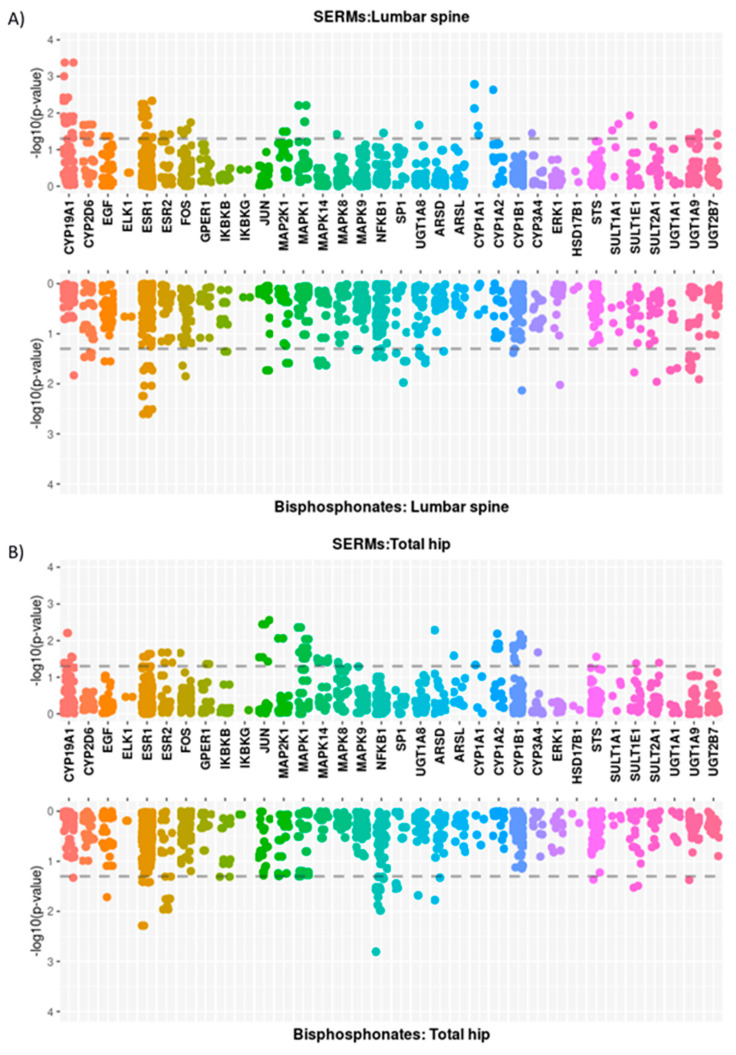
Miami plots showing the *p*-values of the association between oestrogen pathway SNPs and annual BMD change at the spine (**A**) and the hip (**B**). For both regions, the results of the SERM-treated groups (upward) and BP-treated group (downward) are shown.

**Figure 4 pharmaceutics-14-00776-f004:**
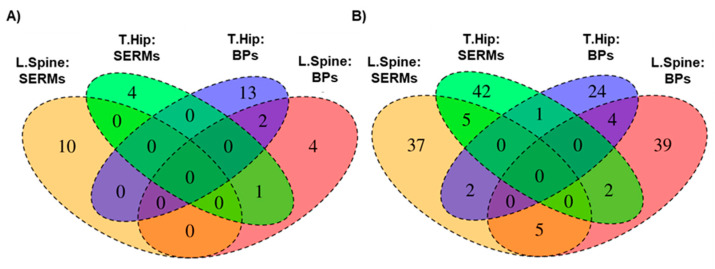
Venn diagrams displaying the intersections between the genome association analyses in the different pathways, the treated groups, and the regions studied (Lumbar spine, L. Spine; and Total hip, T.Hip). (**A**) BP- vs. SERM- treated patients, mevalonate-related genes.; (**B**) BP- vs. SERM-treated patients in the oestrogen-related genes.

**Figure 5 pharmaceutics-14-00776-f005:**
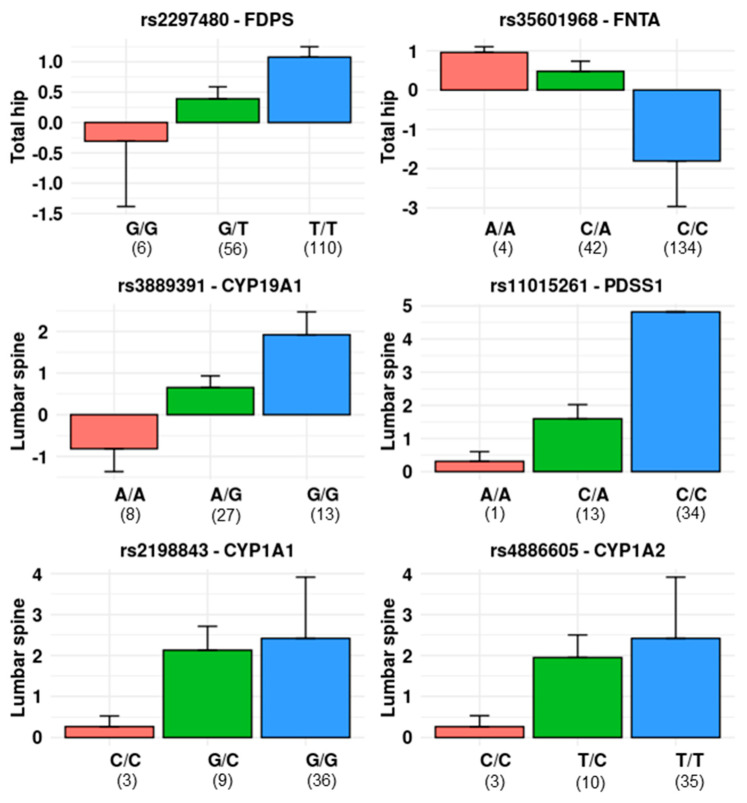
Barplots showing the percent of annual change in BMD (mean and SEM) across different SNP genotypes following bisphosphonate (**upper row** panels) or SERM (**middle** and **lower row** panels). Numbers in parentheses represent the number of women in each group.

**Figure 6 pharmaceutics-14-00776-f006:**
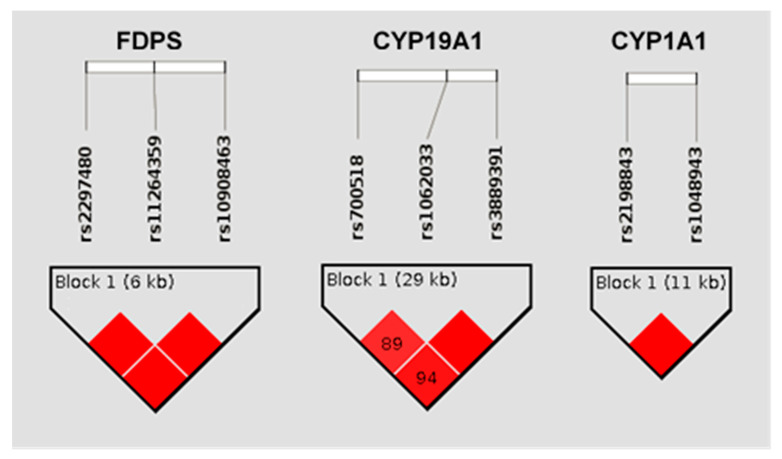
LD patterns of the most significant SNPs of the FDPS, CYP19A1, and CYP1A1 genes. The numbers in each square show between loci LD distances (D′). When no number is shown, D’ = 100. In FDPS and CYP19A1 genes, other variants were included because they were in the same block and also slightly associated with BMD.

**Figure 7 pharmaceutics-14-00776-f007:**
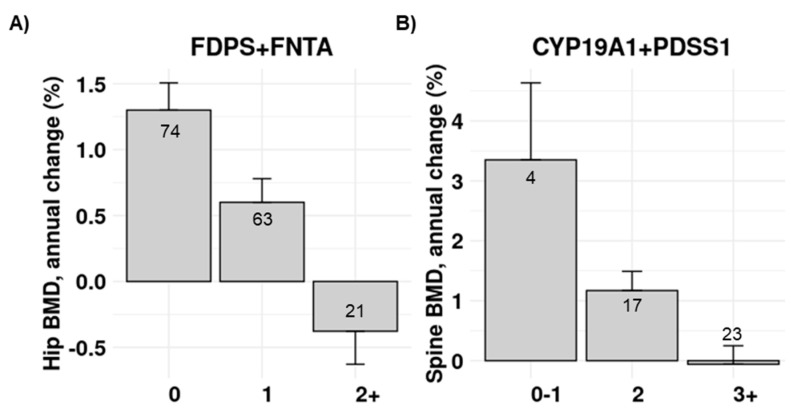
Bar plots showing the average (mean and SEM) changes in BMD across different genotypes combining alleles of two genes: (**A**) annual change in hip BMD after BP therapy across combined FDPS and FNTA genotypes; (**B**) annual change in spine BMD after SERM therapy across combined CYP19A1 and PDSS1 genotypes. Genotypes are divided into groups (*X*-axis) according to the number of risk alleles (that is, alleles associated with smaller drug response). The numbers in the bars represent the number of women in each group.

**Figure 8 pharmaceutics-14-00776-f008:**
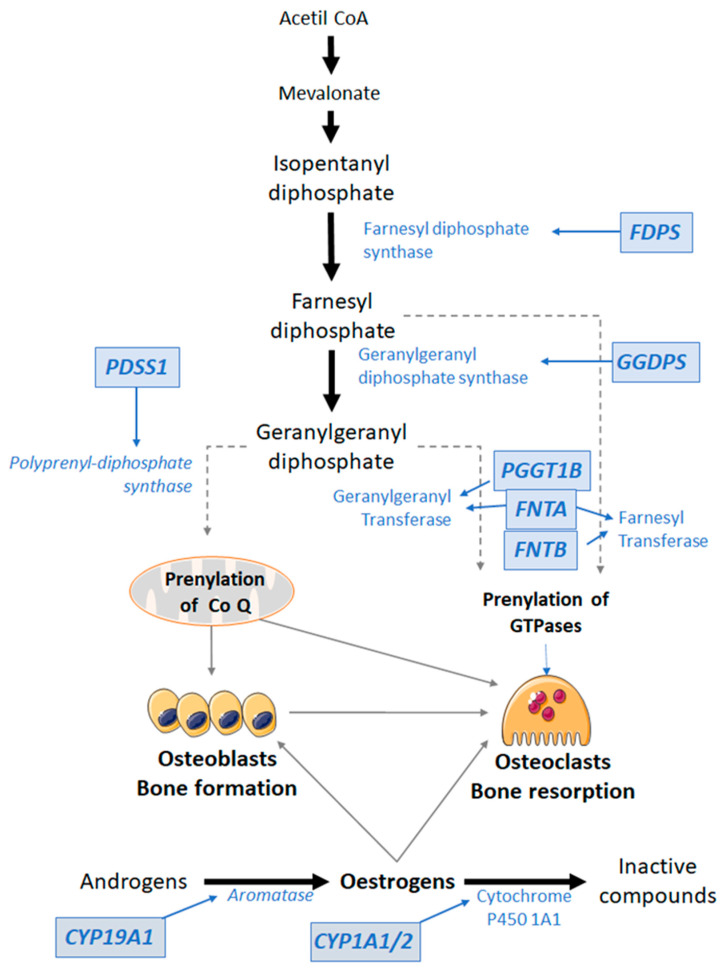
Genes and enzymes associated with antiresorptive drug response in the present study and their relationship with osteoblast and osteoclast activity.

**Figure 9 pharmaceutics-14-00776-f009:**
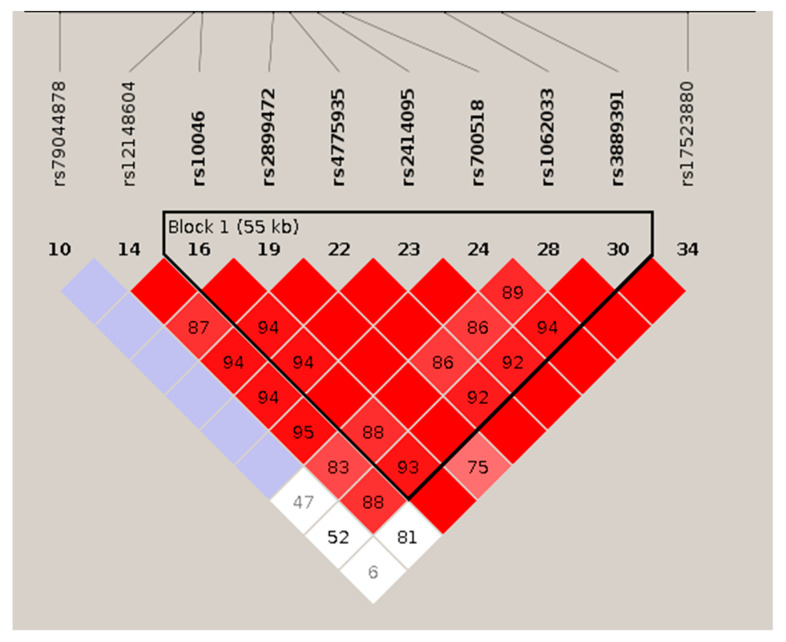
Linkage disequilibrium across the CYP19A1 gene. By using Gabriel’s method, a long 55 Kb haplotypic block with 7 SNPs is revealed.

**Table 1 pharmaceutics-14-00776-t001:** Study population. LS, lumbar spine DXA; TH, total hip DXA.

Patient Characteristics	Bisphosphonate Group (*n* = 192; LS = 154; TH = 180)	SERM Group (*n* = 51; LS = 48; TH = 50)
Age, yr	67 ± 9	59 ± 7
BMI, Kg/m^2^	25.7 ± 4	25.9 ± 3.8
Baseline Spine DXA, g/cm^2^	0.745 ± 0.098	0.757 ± 0.093
Baseline Hip DXA, g/cm^2^	0.733 ± 0.095	0.781 ± 0.085
Alendronate, *n*	143	-
Risedronate, *n*	49	-
Raloxifene, *n*	-	43
Bazedoxifene, *n*	-	8
Between-DXA interval, yr	2.6 ± 0.6	2.6 ± 0.7

**Table 2 pharmaceutics-14-00776-t002:** Association of individual SNPs with drug response.

Probes Group	Region	Treatment	SNPs with *p* < 0.05 (*n*)	Genes with Significantly- Associated SNPs
Mevalonate genes	Lumbar spine	Bisphosphonates	7	FNTB; MVD; PDSS1; PDSS2
		SERMs	10	ACAT1; FNTB; HMGCR; PDSS1; PDSS2; ZMPSTE24
	Total hip	Bisphosphonates	15	FDPS; FNTA; FNTB; IDI1; MVD; MVK; PDSS1; PDSS2
		SERMs	5	FNTB; MVD; MVK; PDSS2
Oestrogen-related genes	Lumbar spine	Bisphosphonates	50	CYP19A1; CYP2D6; EGF; ESR1; FOS; IKBKB; JUN; MAP2K1; MAPK14; MAPK9; NFKB1; SP1; UGT1A8; ARSD; CYP1B1
		SERMs	49	CYP19A1; CYP2D6; EGF; ESR1; ESR2; FOS; MAP2K1; MAPK1; MAPK8; NFKB1; UGT1A8; CYP1A1; CYP1A2; CYP3A4; SULT1A1
	Total hip	Bisphosphonates	31	CYP19A1; EGF; ESR1; ESR2; IKBKB; NFKB1; SP1; UGT1A8; ARSD; STS; SULT1E1; UGT1A9
		SERMs	50	CYP19A1; ESR1; ESR2; FOS; GPER1; JUN; MAP2K1; MAPK1; MAPK14; MAPK8; ARSD; ARSL; CYP1A1; CYP1A2; CYP1B1

**Table 3 pharmaceutics-14-00776-t003:** SNP association with BMD across genes of the mevalonate and oestrogen pathways.

Gene	Number of SNPs	Adjusted *p* Threshold	Bisphosphonates		SERMs	Significant SNPs (*n*)
			Lowest *p*-Value; Spine	Lowest *p*-Value; Hip	Lowest *p*-Value; Spine	
FDPS	4	0.0125	0.0816	0.0041	0.3010	2
FNTA	4	0.0125	0.1570	0.0035	0.2889	1
PDSS1	21	0.0024	0.0038	0.0206	0.0020	1
CYP19A1	42	0.0012	0.0147	0.0469	0.0004	2
CYP1A1	4	0.0125	0.2408	0.2104	0.0016	2
CYP1A2	16	0.0031	0.0835	0.1776	0.0023	1

**Table 4 pharmaceutics-14-00776-t004:** Haplotype-based analysis of the association between genes and drug response.

Gene	Haplotype	β	r^2^	*p*	Frequency	SNPs
CYP19A1	GA	−1.35	0.2393	0.0004	0.447	rs1062033; rs3889391
CYP19A1	GG	0.74	0.0067	0.5801	0.020	rs1062033; rs3889391
CYP19A1	CG	1.26	0.2120	0.0009	0.531	rs1062033; rs3889391
CYP1A1	GC	2.27	0.1483	0.0075	0.053	rs2198843; rs1048943
CYP1A1	GT	0.97	0.0637	0.0869	0.085	rs2198843; rs1048943
CYP1A1	CT	−1.52	0.2000	0.0016	0.861	rs2198843; rs1048943
FDPS	GT	−0.69	0.0473	0.0041	0.197	rs2297480; rs10908463
FDPS	TC	0.63	0.0418	0.0071	0.799	rs2297480; rs10908463

**Table 5 pharmaceutics-14-00776-t005:** Gene-based analysis using MAGMA; *p*-values of the association of BMD changes with genes are shown in each treatment group.

Gene	Bisphosphonates		SERMs
	Spine	Hip	Spine
FDPS	0.1804	0.0111	0.6361
FNTA	0.4662	0.0665	0.6553
PDSS1	0.0277	0.1269	0.1195
CYP19A1	0.9562	0.8625	0.0086
CYP1A1	0.5439	0.6565	0.0055
CYP1A2	0.1211	0.8432	0.1692

**Table 6 pharmaceutics-14-00776-t006:** Association of genes of the mevalonate pathway with the BP-induced changes in hip BMD; results of the linear regression of the most significant SNP in each gene; Both the univariate and multivariate analyses are shown.

Gene	β	r	*p*
FDPS	−0.68 (−1.17 to −0.18)	0.21	0.008
FNTA	−0.93 (−1.48 to −0.38)	0.26	0.001
FDPS & FNTA	-	0.34	0.00009

**Table 7 pharmaceutics-14-00776-t007:** Association of genes of the oestrogen pathway with the SERM-induced changes in spine BMD. Results of the linear regression of the most significant SNP in each gene, including both the univariate and multivariate analyses, are shown.

Gene	β	r	*p*
CYP19A1	−1.32 (−2.06 to −0.583)	0.48	0.001
PDSS1	1.65 (0.69–2.61)	0.47	0.001
CYP1A1	1.35 (0.49 to 2.02)	0.44	0.003
CYP1A2	1.29 (0.44–2.14)	0.43	0.004
CYP19A1,PDSS1,CYP1A1,CYP1A2	-	0.63	0.0004
CYP19A1 & PDSS1	-	0.62	0.0003

## Data Availability

Not applicable.

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
