# Peer review of "Pharmacogenetics of Osteoporosis: A Pathway Analysis of the Genetic Influence on the Effects of Antiresorptive Drugs"

_pharmaceutics, 2022, doi:10.3390/pharmaceutics14040776_

Round 1

Reviewer 1 Report

A nice well designed study  and well illustrated article.

Major comment:

1.The study refers to the effects of anti-resorptive drugs in women with osteoporosis. The authors state that „ therapy -induced  changes in BMD can provide useful information  on drug effectiveness, but usually it takes at least 2 years to develop”.

The authors do not even mention another possibility of assessment of treatment effect by measurement of serum bone turnover markers. This procedure is a non-invasive one, easy to perform and cheap. Concentrations of bone turnover markers after anti-resorptive treatment change very rapidly within approx. 3 months and changes are significant. Already in 2017 Int Osteoporosis Foundation and  Eur Calcif Tissue Soc WG issued recommendations  (Osteoporos Int 2017, 28, 767-774) on the monitoring of effects of bisphosphonate treatment with the use of two reference  bone turnover markers P1NP and CTX recommended earlier by IOF and IFCC. In accordance with these recommendations first measurement should be perfomed at baseline (start of the therapy with BP) and the second one after 3 months. On the basis of the TRIO study decrease of P1NP after 3 months by >38% and decrease of CTX after 3 months by>56%  indicate the beneficial effect of therapy and suport the long term continuation of BP therapy.

From the practical point of view and having in mind patient compliance this approach should not be underestimated. The authors definitely should comment on this at least in the Discussion.

  1. The data on the mean annual changes of BMD show that these changes ranged fron 0.4% to 2.2%. What was the precision of the instrument to perform DXA measurements and which instrument was used for this purpose?

3.What was the blood volume taken from each patient to perform DNA isolation ?

A nice well designed study  and well illustrated article.

Major comment:

1.The study refers to the effects of anti-resorptive drugs in women with osteoporosis. The authors state that „ therapy -induced  changes in BMD can provide useful information  on drug effectiveness, but usually it takes at least 2 years to develop”.

The authors do not even mention another possibility of assessment of treatment effect by measurement of serum bone turnover markers. This procedure is a non-invasive one, easy to perform and cheap. Concentrations of bone turnover markers after anti-resorptive treatment change very rapidly within approx. 3 months and changes are significant. Already in 2017 Int Osteoporosis Foundation and  Eur Calcif Tissue Soc WG issued recommendations  (Osteoporos Int 2017, 28, 767-774) on the monitoring of effects of bisphosphonate treatment with the use of two reference  bone turnover markers P1NP and CTX recommended earlier by IOF and IFCC. In accordance with these recommendations first measurement should be perfomed at baseline (start of the therapy with BP) and the second one after 3 months. On the basis of the TRIO study decrease of P1NP after 3 months by >38% and decrease of CTX after 3 months by>56%  indicate the beneficial effect of therapy and suport the long term continuation of BP therapy.

From the practical point of view and having in mind patient compliance this approach should not be underestimated. The authors definitely should comment on this at least in the Discussion.

  1. The data on the mean annual changes of BMD show that these changes ranged fron 0.4% to 2.2%. What was the precision of the instrument to perform DXA measurements and which instrument was used for this purpose?

3.What was the blood volume taken from each patient to perform DNA isolation ?

Author Response

1.The study refers to the effects of anti-resorptive drugs in women with osteoporosis. The authors state that „ therapy -induced changes in BMD can provide useful information on drug effectiveness, but usually it takes at least 2 years to develop”.

The authors do not even mention another possibility of assessment of treatment effect by measurement of serum bone turnover markers. This procedure is a non-invasive one, easy to perform and cheap. Concentrations of bone turnover markers after anti-resorptive treatment change very rapidly within approx. 3 months and changes are significant. Already in 2017 Int Osteoporosis Foundation and  Eur Calcif Tissue Soc WG issued recommendations  (Osteoporos Int 2017, 28, 767-774) on the monitoring of effects of bisphosphonate treatment with the use of two reference  bone turnover markers P1NP and CTX recommended earlier by IOF and IFCC. In accordance with these recommendations first measurement should be perfomed at baseline (start of the therapy with BP) and the second one after 3 months. On the basis of the TRIO study decrease of P1NP after 3 months by >38% and decrease of CTX after 3 months by>56%  indicate the beneficial effect of therapy and suport the long term continuation of BP therapy.

From the practical point of view and having in mind patient compliance this approach should not be underestimated. The authors definitely should comment on this at least in the Discussion.

Thanks for the comment. We agree that this may be an interesting endpoint in future studies. Now we mention this in Discussion and include two new references about the relationship between bone turnover markers and fracture risk.

2.The data on the mean annual changes of BMD show that these changes ranged fron 0.4% to 2.2%. What was the precision of the instrument to perform DXA measurements and which instrument was used for this purpose?

We now provide the details in Methods. Please, note that we show the mean annual change in BMD, but the actual change over the treatment period was substantially larger, accordingly to the number of years elapsed between the two DXA scans.

3.What was the blood volume taken from each patient to perform DNA isolation?

200 mcl, as it is now stated in Methods

Reviewer 2 Report

Comments and suggestions for authors:

The paper by del Real et al. presents the effects of bisphosphonates and selective oestrogen receptor modulators (SERM) in the treatment of the osteoporosis and its effects on bone mineral density (BMD).  Additionally, the study demonstrates the relationship between the common variants of genes in mevalonate and oestrogen pathways and their influence to drug responses. The study is of high quality, well conducted, meaningful, and deserves to be accepted after some revision performed.

My main suggestion is to re-organized the Discussion section and make it clearer and easier to read, focusing on your results and possible effects.

Figures and Tables and easy to follow and very well organized.

Minor changes:

Line 3: make the title without dividing the word EFFECTS

Line 207: change capital A into a small letter

Line 264: extra bracket

Line 369: add space after 22.1%

Author Response

The paper by del Real et al. presents the effects of bisphosphonates and selective oestrogen receptor modulators (SERM) in the treatment of the osteoporosis and its effects on bone mineral density (BMD).  Additionally, the study demonstrates the relationship between the common variants of genes in mevalonate and oestrogen pathways and their influence to drug responses. The study is of high quality, well conducted, meaningful, and deserves to be accepted after some revision performed.

My main suggestion is to re-organized the Discussion section and make it clearer and easier to read, focusing on your results and possible effects.

Thanks for the comments. We have now included several subheadings in Discussion trying to make it easier to read and follow.

Figures and Tables and easy to follow and very well organized.

Minor changes:

Line 3: make the title without dividing the word EFFECTS

Line 207: change capital A into a small letter

Line 264: extra bracket

Line 369: add space after 22.1%

Thanks for the comments. We have corrected those mistakes.
